# SINGLE SHOT NEURAL ARCHITECTURE SEARCH VIA DIRECT SPARSE OPTIMIZATION

## ABSTRACT

Recently Neural Architecture Search (NAS) has aroused great interest in both academia and industry, however it remains challenging because of its huge and non-continuous search space. Instead of applying evolutionary algorithm or reinforcement learning as previous works, this paper proposes a Direct Sparse Optimization NAS (DSO-NAS) method. In DSO-NAS, we provide a novel model pruning view to NAS problem. Specifically, we start from a completely connected block, and then introduce scaling factors to scale the information flow between operations. Next, we impose sparse regularizations to prune useless connections in the architecture. Lastly, we derive an efficient and theoretically sound optimization method to solve it. Our method enjoys both advantages of differentiability and efficiency, therefore can be directly applied to large datasets like ImageNet. Particularly, on CIFAR-10 dataset, DSO-NAS achieves an average test error 2.84%, while on the ImageNet dataset DSO-NAS achieves 25.4% test error under 600M FLOPs with 8 GPUs in 18 hours.

## 1 INTRODUCTION

With no doubt, Deep Neural Networks (DNN) have been the engines for the AI renaissance in recent years. Dating back to 2012, DNN based methods have refreshed the records for many AI applications, such as image classification (Krizhevsky et al. (2012); Szegedy et al. (2015); He et al. (2016)), speech recognition (Hinton et al. (2012); Graves et al. (2013)) and Go Game (Silver et al. (2016; 2017)). Considering its amazing representation power, DNNs have shifted the paradigm of these applications from manually designing the features and stagewise pipelines to end-to-end learning. Although DNNs have liberated researchers from such feature engineering, another tedious work has emerged – "network engineering". In most cases, the neural networks need to be designed based on the specific tasks, which again leads to endless hyperparameters tuning and trails. Therefore, designing a suitable neural network architecture still requires considerable amounts of expertise and experience.

To democratize the techniques, Neural Architecture Search (NAS) or more broadly, AutoML has been proposed. There are mainly two streams for NAS: The first one is to follow the pioneering work Zoph & Le (2017), which proposed a reinforcement learning algorithm to train a Recurrent Neural Network (RNN) controller that generates coded architectures (Zoph et al. (2018); Pham et al. (2018)). The second one is the evolutionary algorithm, which iteratively evaluates and proposes new models for evaluation (Real et al. (2017); Stanley & Miikkulainen (2002)). Despite their impressive performance, the search processes are incredibly resource-hungry and unpractical for large datasets like ImageNet, though some acceleration methods have been proposed (Zhong et al. (2018); Pham et al. (2018)). Very recently, DARTS (Liu et al. (2018b)) proposed a gradient-based method in which the connections are selected by a softmax classifier. Although DARTS achieves impressive performance with great acceleration, its search space is still limited to fix-length coding and block-sharing search as in previous works.

In this work, we take another view to tackle these problems. We reformulate NAS as pruning the useless connections from a large network which contains the complete network architecture hypothesis space. Thus only one single model is trained and evaluated. Since the network structure is directly optimized during training, we call our method Direct Sparse Optimization NAS (DSO-NAS). We further demonstrate that this sparse regularized problem can be efficiently optimized by

a modified accelerated proximal gradient method opposed to the inefficient reinforcement learning or revolutionary search. Notably, DSO-NAS is much simpler than the existing search methods as it unifies the neural network weight learning and architecture search into one single optimization problem. DSO-NAS does not need any controller (Zoph & Le (2017); Zoph et al. (2018); Pham et al. (2018)) or performance predictor (Liu et al. (2018a)) or relaxation of the search space (Zoph & Le (2017); Zoph et al. (2018); Pham et al. (2018); Liu et al. (2018b)). As a result of the efficiency and simplicity, *DSO-NAS first demonstrate that NAS can be directly applied to large datasets like ImageNet with no block structure sharing.* Our experiments show that DSO-NAS can achieve 2.84% average test error on CIFAR-10, as well as top-1 error 25.4% on ImageNet with FLOPs (the number of multiply-adds) under 600M.

In summary, our contributions can be summarized as follows:

- We propose a novel model pruning formulation for neural architecture search based on sparse optimization. Only one model needs to be trained during the search.

- We propose a theoretically sound optimization method to solve this challenging optimization problem both effectively and efficiently.

- We demonstrate the results of our proposed method are competitive or better than other NAS methods, while significantly simplifying and accelerating the search process.

## 2 RELATED WORKS

In this section, we briefly review two research fields that may be related to our work.

### 2.1 NETWORK PRUNING

Network pruning is a widely used technique for model acceleration and compression. The early works of pruning focus on removing unimportant connections (LeCun et al. (1990); Hassibi & Stork (1993); Han et al. (2015); Guo et al. (2016)). Though connection level pruning can yield effective compression, it is hard to harvest actual computational savings because modern GPU cannot utilize the irregular weights well. To tackle this issue, a significant amount of works on structure pruning have been proposed. For neuron level pruning, several works prune the neurons directly by evaluating the importance of neurons based on specific criteria (Hu et al. (2016); Li et al. (2017); Mariet & Sra (2016); Liu et al. (2017)). More generally, Wen et al. (2016) proposed the sparse structure learning. They adopted group sparsity on multiple structures of networks, including filter shapes, channels and layers. Recently, Huang & Wang (2018) proposed a simpler way for structure pruning. They introduced scaling factors to the outputs of specific structures (neural, groups or block) and apply sparse regularizations on them. After training, structures with zero scaling factors can be safely removed. Compared with (Wen et al. (2016)), the proposed method is more effective and stable. In this work, we extend (Huang & Wang (2018)) into a more general and harder case, neural architecture search.

### 2.2 NEURAL ARCHITECTURE SEARCH

Recently, there has been growing interest in developing methods to generate neural network architecture automatically. One heavily investigated direction is evolutionary algorithm (Meyer-Lee et al.; Miller et al. (1989); Real et al. (2017); Stanley & Miikkulainen (2002)). They designed the modifications like inserting layers, changing filter sizes or adding identity mapping as the mutations in evolution. Not surprisingly, their methods are usually computationally intensive and less practical in large scale. Another popular direction is to utilize reinforcement learning with an RNN agent to design the network architecture. The pioneering work (Zoph & Le (2017)) applies an RNN network as the controller to sequentially decide the type and parameters of layers. Then the controller is trained by RL with the reward designed as the accuracy of the proposed model. Although it achieves remarkable results, it needs 800 GPUs to get such results, which is not affordable for broad applications. Based on this work, several methods have been proposed to accelerate the search process by limiting the search space (Zoph et al. (2018)), early stopping with performance prediction (Zhong et al. (2018)), progressive search (Liu et al. (2018a)) or weight sharing (Pham et al. (2018)). Despite

their success, the aforementioned methods treat the search of network architecture as a black-box optimization problem. Moreover, the search spaces of them are limited due to the fixed-length coding of architecture.

More recently, Ashok et al. (2018); He et al. (2018) integrated reinforcement learning into model compression. For efficient search, the action space is constrained as pruning ratio in He et al. (2018). Though they only applied this method on channel selection, it also can be seen as an architecture search approach. Similarly, Véniat & Denoyer (2018) also constrained the search space by predefining it as a super-network, which is composed of a set of layers connected together. In order to learn budget-aware network structure, they proposed a budget aware objective function and solved it by policy gradient inspired algorithm. Though the building of the search space is similar to our method, the network structure is limited to ResNet fabric and convolutional neural fabrics.

Our most related work is a gradient based method DARTS (Liu et al. (2018b)). In DARTS, a special parameter $a$ is applied on every connection and updated during training process. A Softmax classifier is then applied to select the connection to be used for nodes. However, the search space of DARTS is also limited: every operation can only have exact two inputs; the number of nodes are fixed within a block.

## 3 PROPOSED METHOD

In this section, we will elaborate the details of our proposed method. We will start with the intuition and motivations, then followed by the design of search space and the formulation of our method. Lastly, we will describe the optimization and training details.

### 3.1 MOTIVATIONS

The idea of DSO-NAS follows the observation that the architecture space of neural network (or a micro structure in it) can be represented by a completely connected Directed Acyclic Graph (DAG). Any other architecture in this space can be represented by a sub-graph of it. In other words, a specific architecture can be obtained by selecting a subset of edges and nodes in the full graph. Prior works (Zoph & Le (2017), Liu et al. (2018a), Liu et al. (2018b)) focus on searching the architecture of two types of blocks, convolution block and reduction block. Following the idea of micro structure searching, we adopt the complete graph to represent the search space of an individual block. Then the final network architecture can be represented by a stacking of blocks with residual connections. Fig. 1 illustrates an example DAG of a specific block, whose nodes and edges represent local computation $\mathcal{O}$ and information flow, respectively.

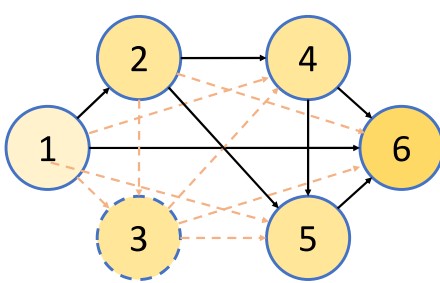

Figure 1: The whole search space can be represented by a completely connected DAG. Here node 1 and 6 are the input and output node, respectively. The dashed line and dashed circle represent that the corresponding connections and nodes are removed. For example, the initial output of node 5 can be calculated by $\mathbf{h}^{(5)} = \mathcal{O}^{(5)}(\sum_{j=1}^{4} \mathbf{h}^{(j)})$, while it becomes $\mathbf{h}^{(5)} = \mathcal{O}^{(5)}(\mathbf{h}^{(2)} + \mathbf{h}^{(4)})$ for the pruned sub-graph.

For a DAG with $T$ nodes, the output of $i$th node $\mathbf{h}^{(i)}$ can be calculated by transforming the sum of all the outputs of the predecessors, $\mathbf{h}^{(j)}, j < i$, by the local operation $\mathcal{O}^{(i)}$, namely:

$$\mathbf{h}^{(i)} = \mathcal{O}^{(i)}(\sum_{j=1}^{i-1} \mathbf{h}^{(j)}). \quad (1)$$

Then the structure search problem can be reformulated as an edge pruning problem. In the search procedure, we remove useless edges and nodes in the full DAG, leaving the most important structures. To achieve this goal, we apply scaling factors on every edge to scale the output of each node. Then Eqn. 1 can be modified to:

$$\mathbf{h}^{(i)} = \mathcal{O}^{(i)}(\sum_{j=1}^{i-1} \lambda_{(j)}^{(i)} \mathbf{h}^{(j)}), \quad (2)$$

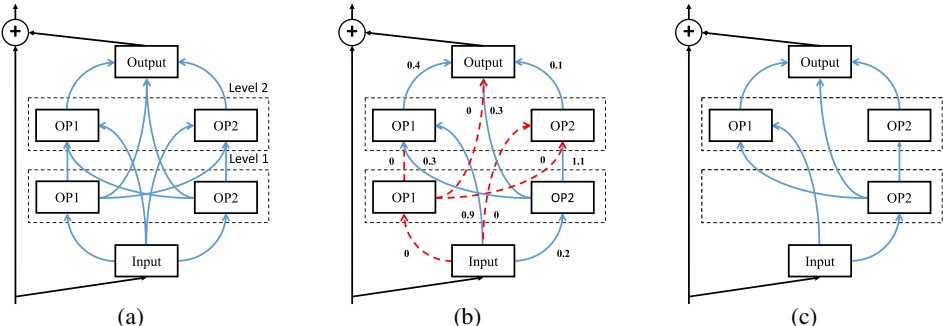

Figure 2: An example of search block, which has two levels with two operations: (a) The completely connected block. (b) In the search process, we jointly optimize the weights of neural network and the $\lambda$ associated with each edge. (c) The final model after removing useless connections and operations.

where $\lambda_{(j)}^{(i)}$ is the scaling factor applied on the information flow from node $j$ to $i$. Then we apply sparse regularizations on the scaling parameters to force some of them to be zero in the search. Intuitively, if $\lambda_{(j)}^{(i)}$ is zero, the corresponding edge can be removed safely and isolated nodes can also be pruned as no contribution is made.

## 3.2 SEARCH SPACE

DSO-NAS can search the structure of each building block in DNN, and then share it for all the blocks in the DNN, just as all previous works did. It can also directly search the whole network structure without block structure sharing, while still keeping a competitive searching time. In the following, we will discuss the search space of each individual block first, and then specify the search space of macro-structure.

A block consists of $M$ sequential levels which are composed of $N$ different kinds of operations. In each block, every operation has connections with all the operations in the former levels and the input of the block. Also, the output of the block is connected with all operations in the block. Then for each connection, we scale its output by a multiplier $\lambda$, and impose a sparse regularization on it. After optimization, the final architecture is generated by pruning the connections whose corresponding $\boldsymbol{\lambda}$ are zero and all isolated operations. The procedure of block search is illustrated in Fig. 2. Formally, the output of the $j$-th operation in the $i$-th layer of the $b$-th block $\mathbf{h}_{(b,i,j)}$ is computed as:

$$\mathbf{h}_{(b,i,j)} = \mathcal{O}_{(b,i,j)}(\sum_{m=1}^{i-1}\sum_{n=1}^{N}\lambda_{(b,m,n)}^{(i,j)}\mathbf{h}_{(b,m,n)} + \lambda_{(b,0,0)}^{(i,j)}\mathbf{O}_{(b-1)}), \tag{3}$$

where $\mathcal{O}_{(b,i,j)}$ is the transformation of the $j$-th operation in the $i$-th layer of the $b$-th block, $\lambda_{(b,m,n)}^{(i,j)}$ is the scaling factor from node $\mathbf{h}_{(b,m,n)}$ to $\mathbf{h}_{(b,i,j)}$, and $\mathbf{O}_{(b-1)}$ is the output of the $(b-1)$-th block. Here we denote $\mathbf{h}_{(b,0,0)} = \mathbf{O}_{(b-1)}$ as the input node and $\mathbf{h}_{(b,M+1,0)} = \mathbf{O}_{(b)}$ as the output node of the $b$-th block, respectively. The operation in the $m$-th layer may have $(m-1)N + 1$ inputs. Note that the connections between operations and the output of block are also learnable. The output of the $b$-th block $\mathbf{O}_{(b)}$ is obtained by applying a reduction operation (concatenation followed by a convolution with kernel size 1×1) $\mathcal{R}$ to all the nodes that have contribution to the output:

$$\mathbf{O}_{(b)} = \mathcal{R}([\lambda_{(b,1,1)}^{(M+1,0)}\mathbf{h}_{(b,1,1)},\lambda_{(b,1,2)}^{(M+1,0)}\mathbf{h}_{(b,1,2)}, ..., \lambda_{(b,m,n)}^{(M+1,0)}\mathbf{h}_{(b,m,n)}, ...\lambda_{(b,M,N)}^{(M+1,0)}\mathbf{h}_{(b,M,N)}])$$
$$+ \mathbf{O}_{(b-1)}, m \in [1,M], n \in [1,N] \tag{4}$$

where identity mapping is applied in case all operations are pruned.

The structure of whole network is shown in Fig. 3: a network consists of $S$ stages with $B$ convolution blocks in every stage. Reduction block is located at the end of each stage except the last stage. We

try two search spaces: (1) the share search space where $\boldsymbol{\lambda}$ is shared among blocks. (2) the full search space where $\boldsymbol{\lambda}$ in different blocks are updated independently.

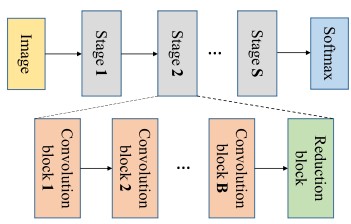

Figure 3: The structure of network.

We use the Conv-Bn-ReLU order for convolutional operations and adopt following four kinds of operations in convolution block following Pham et al. (2018):

- Separable convolution with kernel $3\times3$
- Separable convolution with kernel $5\times5$
- Average pooling with kernel $3\times3$
- Max pooling with kernel $3\times3$

As for reduction block, we simply use convolutions with kernel size $1\times1$ and $3\times3$, and apply them with a stride of 2 to reduce the size of feature map and double the number of filters. The output of reduction block is the sum of these two convolutions.

The task of searching blocks therefore reduces to learning $\boldsymbol{\lambda}$ on every edge, which can be formulated as:

$$\min_{\mathbf{W},\boldsymbol{\lambda}} \frac{1}{K} \sum_{i=1}^{K} \mathcal{L}(\mathbf{y}_i, Net(\mathbf{x}_i, \mathbf{W}, \boldsymbol{\lambda})) + \delta\|\mathbf{W}\|_F^2 + \gamma\|\boldsymbol{\lambda}\|_1, \quad (5)$$

where $\mathbf{x}_i$ and $\mathbf{y}_i$ are input data and label respectively, $K$ denotes the number of training samples, $\mathbf{W}$ represents the weights of network. $\delta$ and $\gamma$ represent the weights of regularizations, respectively.

### 3.3 Optimization and Training

The sparse regularization of $\boldsymbol{\lambda}$ induces great difficulties in optimization, especially in the stochastic setting in DNN. Though heuristic thresholding could work, the optimization is unstable and without theoretical analysis. Fortunately, a recently proposed method Sparse Structure Selection (SSS) (Huang & Wang (2018)) solved this challenging problem by modifying a theoretically sound optimization method Accelerated Proximal Gradient (APG) method, it reformulates the original APG to avoid redundant forward and backward pass in calculating the gradients:

$$\mathbf{z}_{(t)} = \boldsymbol{\lambda}_{(t-1)} - \eta_{(t)} \nabla \mathcal{G}(\boldsymbol{\lambda}_{(t-1)}) \quad (6)$$

$$\mathbf{v}_{(t)} = S_{\eta_{(t)}\gamma}(\mathbf{z}_{(t)}) - \boldsymbol{\lambda}_{(t-1)} + \mu_{(t-1)}\mathbf{v}_{(t-1)} \quad (7)$$

$$\boldsymbol{\lambda}_{(t)} = \mathcal{S}_{\eta_{(t)}\gamma}(\mathbf{z}_{(t)}) + \mu_{(t)}\mathbf{v}_{(t)} \quad (8)$$

where $t$ is the number of iterations, $\mathcal{S}_{\eta_{(t)}\gamma}$ represents the soft-threshold operator as $\mathcal{S}_\alpha(\mathbf{z})_i = \mathrm{sign}(z_i)(|z_i| - \alpha)_+$, $\eta_{(t)}$ represents gradient step size and $\mu$ is the momentum. In (Huang & Wang (2018)), the authors named it as APG-NAG. The weights $\mathbf{W}$ and $\boldsymbol{\lambda}$ are updated using NAG and APG-NAG jointly on the same training set. However, APG-NAG cannot be directly applied in our algorithm since DNN usually overfits the training data in some degree. Different from pruning, whose the search space is usually quite limited, the search space in NAS is much more diverse and huge. If the structure is learned on such overfitting model, it will generalize badly on the test set.

To avoid this problem, we divide training data into two parts and update $\mathbf{W}$ and $\boldsymbol{\lambda}$ in two separate sets. This configuration guarantees that the $\boldsymbol{\lambda}$ (i.e. network structure) is learned on a different subset of training data which is not seen during the learning of $\mathbf{W}$. Therefore, the sample distribution in the structure learning is more similar to that in testing, which may lead to better performance.

### 3.4 Incorporating Different Budgets

Hand-crafted network usually incorporates many domain knowledge. For example, as highlighted in (Ma et al. (2018)), memory access may be the bottleneck for lightweight network on GPU because the use of separable convolution. Our method can easily consider these priors in the search by adaptively adjusting $\gamma$ for each connection.

The first example is to balance the FLOPs for each block. As indicated in (Jastrzebski et al. (2018)), most intense changes of the main branch flow of ResNet are concentrated after reduction block. In

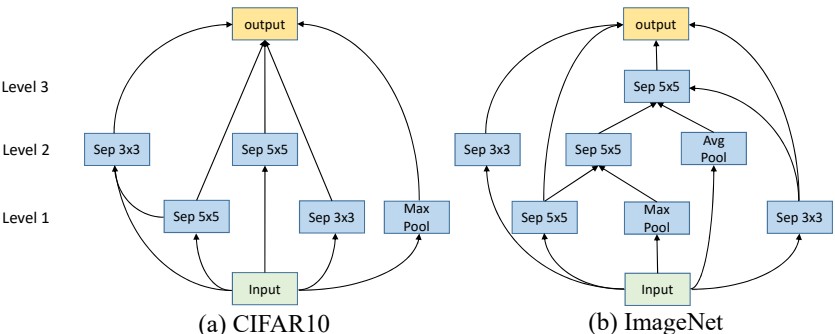

Figure 4: Block structures learned on different datasets.

our experiments, we empirically find that the complexity of the block after each reduction block is much higher than the others' if all $\gamma$ are fixed. Consequently, to balance the FLOPs among different blocks, we adjust the regularization weight for $\boldsymbol{\lambda}$, namely $\gamma^t$ at iteration $t$ adaptively according to the FLOPs of the block:

$$\gamma^t = \frac{\text{FLOPs}^t}{\text{FLOPs}_{block}}\gamma, \tag{9}$$

where $\text{FLOPs}_{block}$ represents the FLOPs of the completely connected block and $\text{FLOPs}^t$, which can be calculated based on $\boldsymbol{\lambda}$, represents the FLOPs of the kept operations at iteration $t$ for a block. Using this simple strategy, we can smooth the distribution of FLOPs by penalizing $\boldsymbol{\lambda}$ according to the FLOPs of the block. We call this method *Adaptive FLOPs* in the following.

The second example is to incorporate specific computation budget such as Memory Access Cost (MAC). Similarly, the $\gamma$ applied on the $n$-th operation in $m$-th level at iteration $t$ is calculated by:

$$\gamma^t_{(m,n)} = \frac{\text{MAC}^t_{(m,n)}}{\text{MAC}_{max}}\gamma, \tag{10}$$

where $\text{MAC}^t_{(m,n)}$ represents the MAC of the $n$-th operation in the $m$ level, and $\text{MAC}_{max}$ represents the maximum MAC in the network. Using this strategy, DSO-NAS can generate architectures with better performance under the same budget of MAC. We call this method *Adaptive MAC* in the following.

## 4 EXPERIMENTS

In this section, we first introduce the implementation details of our method, then followed by the results of classification task on CIFAR-10 and ImageNet datasets. At last, we analyze each design component of our method in detail.

### 4.1 IMPLEMENTATION DETAILS

The pipeline of our method consists of three stages:

1. Training the completely connected network for several epochs to get a good weights initialization.

2. Searching network architecture from the pretrained model.

3. Re-training the final architecture from scratch and evaluating on test dataset.

In the first two stages, the scaling parameters in batch normalization layers are fixed to one to prevent affecting the learning of $\boldsymbol{\lambda}$. After step two, we adjust the number of filters in each operation by a global width multiplier to satisfy the computation budget, and then train the network from scratch as done in (Pham et al. (2018)).

Table 1: Comparison with state-of-the-art NAS methods on CIFAR-10.

| Architecture | Test Error | Params(M) | Search Cost (GPU days) | Search Method |
|---|---|---|---|---|
| DenseNet | 3.46 | 25.6 | - | manual |
| NASNet-A + c/o (Zoph et al. (2018)) | 2.65 | 3.3 | 1800 | RL |
| AmoebaNet-A (Real et al. (2018)) | 3.34 | 3.2 | 3150 | evolution |
| AmoebaNet-B + c/o (Real et al. (2018)) | 2.55 | 2.8 | 3150 | evolution |
| PNAS (Liu et al. (2018a)) | 3.41 | 3.2 | 150 | SMBO |
| ENAS + c/o (Pham et al. (2018)) | 2.89 | 4.6 | 0.5 | RL |
| DARTS(1st order) + c/o (Liu et al. (2018b)) | 2.94 | 2.9 | 1.5 | gradient |
| DARTS(2nd order) + c/o (Liu et al. (2018b)) | 2.83 | 3.4 | 4 | gradient |
| DSO-NAS-share+c/o | $2.84 \pm 0.07$ | 3.0 | 1 | gradient |
| DSO-NAS-full+c/o | $2.95 \pm 0.12$ | 3.0 | 1 | gradient |
| random-share + c/o | $3.58 \pm 0.21$ | $3.4 \pm 0.1$ | - | - |
| random-full + c/o | $3.52 \pm 0.19$ | $3.5 \pm 0.1$ | - | - |

For benchmark, we test our algorithm on two standard datasets, CIFAR-10 (Krizhevsky & Hinton (2009)) and ImageNet LSVRC 2012 (Russakovsky et al. (2015)). We denote the model searched with and without block structure sharing as *DSO-NAS-share* and *DSO-NAS-full*, respectively. In each block, we set the number of levels $M = 4$, the number of operations $N = 4$ as four kinds of operations are applied for both CIFAR and ImageNet experiments indicated in Sec. 3.2. For the hyper-parameters of optimization algorithm and weight initialization, we follow the setting of (Huang & Wang (2018)). All the experiments are conducted in MXNet (Chen et al. (2015)) with NVIDIA GTX 1080Ti GPUs. We will release our codes if the paper is accepted.

## 4.2 CIFAR

The CIFAR-10 dataset consists of 50000 training images and 10000 testing images. As described in Sec.3.3, we divide the training data into two parts: 25000 for training of weights and rest 25000 for structure. During training, standard data pre-processing and augmentation techniques (Xie et al. (2017)) are adopted. The mini-batch size is 128 on 2 GPUs and weight decay is set to 3e-4. Firstly, we pre-train the full network for 120 epochs, and then search network architecture until convergence (about 120 epochs). Constant learning rate 0.1 is applied in both pre-training and search stages. The network adopted in CIFAR-10 experiments consists of three stages, and each stage has eight convolution blocks and one reduction block. Adaptive FLOPs (see section 3.3) is applied during the search.

After the search, we train the final model from scratch with the same setting of (Pham et al. (2018)). The searched models are trained for 630 epochs with NAG, where we change the learning rate following the cosine scheduler with $l_{max} = 0.05, l_{min} = 0.001, T_0 = 10, T_{mul} = 2$ (Loshchilov & Hutter (2017)). Additional improvements including dropout (Srivastava et al. (2014)) with probability 0.6, cutout (DeVries & Taylor (2017)) with size 16, drop path (Larsson et al. (2016)) with probability 0.5, auxiliary towers located at the end of second stage (Lee et al. (2015)) with weight 0.4.

Table 1 shows the performance of our searched models, including DSO-NAS-full and DSO-NAS-share, where "c/o" represents evaluate model with cutout technique. We report the mean and standard deviation of five independent runs. Due to limited space, we only show the block structure of DSO-NAS-share in Fig.4(a). We also compare the simplest yet still effective baseline – random structure, both our DSO-NAS-share and DSO-NAS-full yield much better performance with less parameters. Comparing with other state-of-the-art methods, our method demonstrates competitive results with similar or less parameters while costing only one GPU day.

## 4.3 ILSVRC 2012

In the ILSVRC 2012 experiments, we conduct data augmentation based on the publicly available implementation of 'fb.resnet'[1]. Since this dataset is much larger than CIFAR-10, the training dataset is divided into two parts: $4/5$ for training weights and $1/5$ for training structure. In the pre-training stage, we train the whole network for 30 epochs with constant learning rate 0.1, weight decay $4 \times 10^{-5}$. The mini-batch size is 256 on 8 GPUs. The same setting is adopted in the search stage, which costs about 0.75 days with 8 GPUs.

After the search, we train the final model from scratch using NAG for 240 epochs, with batch size 1024 on 8 GPUs. We set weight decay to $4 \times 10^{-5}$ and adopt linear-decay learning rate schedule (linearly decreased from 0.5 to 0). Label smoothing (Szegedy et al. (2016)) and auxiliary loss (Lee et al. (2015)) are used during training. There are four stages in the ImageNet network, and the number of convolution blocks in these four stages is 2, 2, 13, 6, respectively. We first transfer the block structure searched on CIFAR-10. We also directly search the network architecture on ImageNet. The final structure generated by DSO-NAS-share is shown in Fig.4(b). The quantitative results for ImageNet are shown in Table 2, where results with * are obtained by transferring the generated CIFAR-10 blocks to ImageNet.

Table 2: Comparison with state-of-the-art image classifiers on ImageNet

| Architecture | Top-1/5 | Params(M) | FLOPS(M) | Search Cost (GPU days) |
|---|---|---|---|---|
| Inception-v1 (Szegedy et al. (2015)) | 30.2/10.1 | 6.6 | 1448 | - |
| MobileNet (Howard et al. (2017)) | 29.4/10.5 | 4.2 | 569 | - |
| ShuffleNet-v1 2x (Zhang et al. (2018)) | 26.3/10.2 | 5 | 524 | - |
| ShuffleNet-v2 2x (Ma et al. (2018)) | 25.1/- | 5 | 591 | - |
| NASNet-A* (Zoph et al. (2018)) | 26.0/8.4 | 5.3 | 564 | 1800 |
| AmoebaNet-C* (Real et al. (2018)) | 24.3/7.6 | 6.4 | 570 | 3150 |
| PNAS* (Liu et al. (2018a)) | 25.8/8.1 | 5.1 | 588 | 150 |
| DARTS* (Liu et al. (2018b)) | 26.9/9.0 | 4.9 | 595 | 4 |
| OSNAS (Bender et al. (2018)) | 25.8/- | 5.1 | - | - |
| MNAS-92 (Tan et al. (2018)) | 25.2/8.0 | 4.4 | 388 | - |
| DSO-NAS* | 26.2/8.6 | 4.7 | 571 | 1 |
| DSO-NAS-full | 25.7/8.1 | 4.6 | 608 | 6 |
| DSO-NAS-share | 25.4/8.4 | 4.8 | 586 | 6 |

It is notable that given similar computation budget, DSO-NAS achieves competitive or better performance than other state-of-the-art methods with less search cost, except for MnasNet whose search space is carefully designed and different from other NAS methods. The block structure transferred from CIFAR-10 dataset also achieves impressive performance, proving the generalization capability of the searched architecture. Moreover, directly searching on target dataset (ImageNet) brings additional improvements.

## 4.4 ABLATION STUDY

In this section, we present some ablation analyses on our method to illustrate the effectiveness and necessity of each component.

### 4.4.1 THE EFFECTIVENESS OF BUDGET AWARE SEARCH

With adaptive FLOPs technique, the weight of sparse regularization for each block will be changed adaptively according to Eqn. 9. We first show the distribution of FLOPs among different blocks in Fig. 5(a). This strategy can prevent some blocks from being pruned entirely as expected. We also show the error rates of different settings in Fig. 5(b) and Fig. 5(c).

---

[1] https://github.com/facebook/fb.resnet.torch

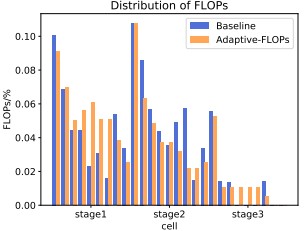
(a) Distribution of FLOPs

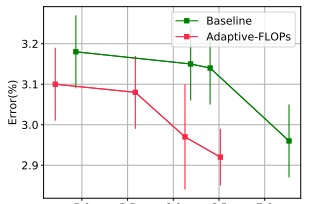
(b) Err./FLOPs for adaptive FLOPs

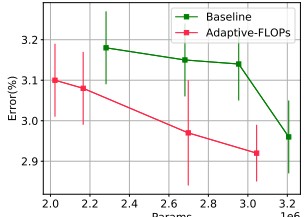
(c) Err./Params. for adaptive FLOPs

Figure 5: Performance of adaptive FLOPs techniques.

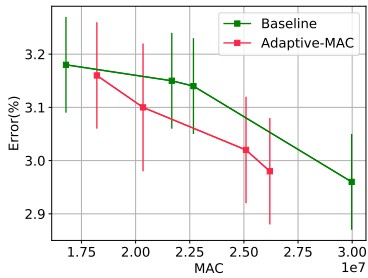

Figure 6: Performance of adaptive MAC technique

It is clear that the networks searched with adaptive FLOPs technique are consistently better than the ones without under the same total FLOPs or parameters.

DSO-NAS can also search for architecture based on certain computational target, such as MAC discussed in Sec. 3.4. The results are shown in Fig. 6. It is obvious to see that DSO-NAS can generate architecture with higher accuracy under certain MAC budget, proving the effectiveness of adaptive MAC technique. The method can similarly be applied to optimize many other computation budgets of interest, which we leave for further study.

### 4.4.2 OTHER FACTORS FOR SEARCHING ARCHITECTURE

We conduct experiments on different settings of our proposed architecture search method to justify the need of each component we designed. The results are shown in Table 3.

Table 3: Comparison of different settings on CIFAR-10 dataset.

| Search space | Split training | Pre-train model | Ratio of W&S | Params(M) | Test Error |
|:---:|:---:|:---:|:---:|:---:|:---:|
| full | ✓ |   | 1:1 | 2.9 | $3.26 \pm 0.08$ |
| full | ✓ | ✓ | 1:1 | 3.0 | $3.02 \pm 0.09$ |
| full | ✓ | ✓ | 4:1 | 2.9 | $3.05 \pm 0.09$ |
| full |   | ✓ | - | 3.0 | $3.20 \pm 0.10$ |
| share | ✓ |   | 1:1 | 3.0 | $3.07 \pm 0.08$ |
| share | ✓ | ✓ | 1:1 | 3.0 | $2.86 \pm 0.09$ |
| share | ✓ | ✓ | 4:1 | 2.9 | $2.89 \pm 0.06$ |
| share |   | ✓ | - | 3.0 | $3.14 \pm 0.06$ |

"Pre-train model" means whether we conduct step one in Sec. 4, while "Split training" means whether to split the whole training set into two sets for weight and structure learning separately. The Ratio of W&S means the ratio of training sample for weight learning and structure learning. As for the ratio of $x : y$, we update weight for $x$ times and update $\lambda$ for $y$ times for every $x + y$ iterations. Note that we only pre-train the model on the weight learning set.

It is notable that the use of a separate set for structure learning plays an important role to prevent overfitting training data, and improve the performance by 0.2%. The ratio of these two sets has minor influence. Besides, a good initialization of weight is also crucial as random initialization of weights may lead to another 0.2% drop on accuracy under that same parameter budgets.

### 5 CONCLUSIONS AND FUTURE WORK

Neural Architecture Search has been the core technology for realizing AutoML. In this paper, we have proposed a Direct Sparse Optimization method for NAS. Our method is appealing to both

academic research and industrial practice in two aspects: First, our unified weight and structure learning method is fully differentiable in contrast to most previous works. It provides a novel model pruning view to the NAS problem. Second, the induced optimization method is both efficient and effective. We have demonstrated state-of-the-art performance on both CIFAR and ILSVRC2012 image classification datasets, with affordable cost (single machine in one day).

In the future, we would like to incorporate hardware features for network co-design, since the actual running speed of the same network may highly vary across different hardware because of cache size, memory bandwidth, etc. We believe our proposed DSO-NAS opens a new direction to pursue such objective. It could push a further step to AutoML for everyone's use.

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
