# OpenReview forum: "Single Shot Neural Architecture Search Via Direct Sparse Optimization"
_ICLR.cc/2019/Conference_

### Official Review · AnonReviewer1 · 2018-10-30

**Rating:** 7
**Confidence:** 3

**Review:**

The authors present an architecture search method where connections are removed with sparse regularization. It produces good network blocks relatively quickly that perform well on CIFAR/ImageNet.

There are a few grammatical/spelling errors that need ironing out.

e.g. "In specific" --> "Specifically" in the abstract, "computational budge" -> "budget" (page 6) etc.

A few (roughly chronological comments).

- Pioneering work is not necessarily equivalent to "using all the GPUs"

- There are better words than "decent" to describe the performance of DARTS, as it's very similar to the results in this work!

- From figure 2 it's not clear why all non-zero connections in (b) are then equally weighted in (c). Would keeping the non-zero weightings be at all helpful?

-  Why have you chosen the 4 operations at the bottom of page 4? It appears to be a subset of those used in DARTS.

- How do you specifically encode the number of surviving connections? Is it entirely dependent on budget?

- You should add DARTS 1st order to table 1.

- Measuring in GPU days is only meaningful if you use the same GPU make for every experiment. Which did you use?

- The ablation study is good, and the results are impressive.

I propose a marginal acceptance for this paper as it produces impressive results in what appears to be a short amount of search time. However, the implementation details are hazy, and some design choices (which operations, hyperparameters etc.) aren't well justified.

------------
UPDATE: Score changed based on author resposne
------------

---

> ### Author Response · Authors · 2018-11-26
> **Response**
>
> Thanks for your thoughtful review. We have given serious considerations of your concerns and revise our manuscript to accommodate your suggestions. Please see the details below.
>
>
> Q1: “There are a few grammatical/spelling errors that need ironing out.”
> A1: We have fixed the typos and grammatical errors in the revision.
>
> Q2: “Pioneering work is not necessarily equivalent to "using all the GPUs"”
> A2: This claim is indeed not accurate we have delete this claim in the revision.
>
> Q3: “There are better words than "decent" to describe the performance of DARTS, as it's very similar to the results in this work!”
> A3: We have changed the word to “impressive” in the revision. However, DSO-NAS indeed outperforms DARTS on ImageNet dataset as illustrated in Table2.
>
> Q4: “From figure 2 it's not clear why all non-zero connections in (b) are then equally weighted in (c). Would keeping the non-zero weightings be at all helpful?”
> A4: In the search stage, the scaling factors are only used to indicate which operators should be pruned. The value of scaling factors do not represent the importances of kept operators since they can be merged into the weights of convolution.
> We also add experiments in CIFAR-10 to compare the performance between keeping the non-zero weightings and equal weightings. The result shows that both of them yield similar performances.
> ---------------------------------------------------------------------------------------------------------------------------
> Architecture         	                    params(M)                   	test error
> ---------------------------------------------------------------------------------------------------------------------------
> DSO-NAS-share+c/o                          3.0                        	2.84
> ---------------------------------------------------------------------------------------------------------------------------
> DSO-NAS-share+c/o+k/w                  3.0                              2.88
> ---------------------------------------------------------------------------------------------------------------------------
> DSO-NAS-full+c/o                              3.0                        	2.95
> ---------------------------------------------------------------------------------------------------------------------------
> DSO-NAS-full+c/o+k/w                      3.0                        	2.96
> ---------------------------------------------------------------------------------------------------------------------------
> where “c/o” represents that training the searched architectures with cutout and “k/w” represents keeping the non-zero weightings in the architectures.
>
> Q5: “Why have you chosen the 4 operations at the bottom of page 4?”
> A5: These four operations were used by ENAS and commonly included in the search space of most NAS papers.
>
> Q6: “How do you specifically encode the number of surviving connections?”
> A6: We don’t directly encode the number of surviving connections. Instead, the number of surviving connections is determined by the weight for L1 regularization, which can be incorporated with certain budget.
>
> Q7: “Measuring in GPU days is only meaningful if you use the same GPU make for every experiment. Which did you use?”
> A7: All of our experiments were conducted by NVIDIA GTX 1080Ti GPU, which was also used by ENAS and DARTS. We have added it in the paper.

---

> > ### Comment · AnonReviewer1 · 2018-11-29
> > **Reply to the authors**
> >
> > Thank you for your detailed response.
> >
> > I am satisfied with your answers to my questions, and I think this work deserves to be seen by the wider community as a good comparison point to other architecture search schemes, such as DARTS.
> >
> > As such, I will bump my score up to 7, but I implore the authors to do another rewrite as the grammar still leaves much to be desired; you don't want to put people off reading your work because of something so trivial.

---

### Official Review · AnonReviewer3 · 2018-11-02
**If we focus on the balance between the classification accuracy and computational efficiency, the proposed method is promising**

**Rating:** 6
**Confidence:** 3

**Review:**


- Summary
This paper proposes a neural architecture search method based on a direct sparse optimization, where the proposed method provides a novel model pruning view to the neural architecture search problem. Specifically, the proposed method introduces scaling factors to connections between operations, and impose sparse regularizations to prune useless connections in the network. The proposed method is evaluated on CIFAR-10 and ImageNet dataset.

- Pros
  - The proposed method shows competitive or better performance than existing neural architecture search methods.
  - The experiments are conducted thoroughly in the CIFAR-10 and ImageNet. The selection of the datasets is appropriate. Also, the selection of the methods to be compared is appropriate.
  - The effect of each proposed technique is appropriately evaluated.

- Cons
  - The search space of the proposed method, such as the number of operations in the convolution block, is limited.
  - The proposed method does not outperform the existing state-of-the-art methods in terms of classification accuracy.
  - The technical contribution of the proposed method is not high, because the architecture space of neural network is similar to the prior works.

Overall, if we focus on the balance between the classification accuracy and computational efficiency, the proposed method is promising.

---

> ### Author Response · Authors · 2018-11-26
> **Response**
>
> Thanks for pointing out the pros and cons of our method. We address your concerns as follows:
>
> Q1. “The search space of the proposed method, such as the number of operations in the convolution block, is limited.”
> A1: First, the size of search space is not determined by the number of operations but the number of connections. The search space of our method is different from exiting NAS methods in that the number of input of certain operation is not limited.
>
> Second, the search space without block share is even much larger than existing NAS methods.
>
> Third, we can trivially extend our DSO-NAS to accommodate more operations such as dilated conv like our ongoing experiments on PASCAL VOC semantic segmentation task, we extend our search space to accommodate 3x3 and 5x5 separable convolution with dilated = 2. The following table shows the performance of our model on the PASCAL VOC 2012 semantic segmentation task, where DSO-NAS-cls represents the architecture searched on ImageNet with block structure sharing and DSO-NAS-seg represents the architecture searched on PASCAL VOC segmentation task.
> ---------------------------------------------------------------------------------------------------------------------------
> Architecture                                         mIOU                     Params(M)                     FLOPS(B)
> ---------------------------------------------------------------------------------------------------------------------------
> DSO-NAS-cls                                            72.1                            6.5                               13.0
> ---------------------------------------------------------------------------------------------------------------------------
> DSO-NAS-seg(more operations)         72.7                            6.7                               13.2
> ---------------------------------------------------------------------------------------------------------------------------
> We combine DSO-NAS with Deeplab v3 and search for the architecture of feature extractor with block sharing. All above models have been pre-trained on ImageNet classification task first. It’s notable that the architecture searched on semantic segmentation task with additional operations achieve better performance in our preliminary experiment, indicating that our DSO-NAS is capable to incorporate additional operations. We will present the full experiments of semantic segmentation in the future revision.
>
> Q2: “The technical contribution of the proposed method is not high, because the architecture space of neural network is similar to the prior works.”
> A2: Please refer to Q1. Moreover, we never claim the main contribution of our work lies in augmenting the search space. And in fact, most existing NAS papers share the same architecture search space, the main differences between them is the search strategy. We believe that judging the novelty of a NAS paper solely by its architecture space is unfair.

---

### Official Review · AnonReviewer2 · 2018-11-03

**Rating:** 6
**Confidence:** 4

**Review:**

Summary:
This paper proposes Direct Sparse Optimization (DSO)-NAS, which is a method to obtain neural architectures on specific problems, at a reasonable computational cost.

The main idea is to treat all architectures as a Directed Acyclic Graph (DAG), where each architecture is realized by a subgraph. All architectures in the search space thus share their weights, like ENAS (Pham et al 2018) and DARTS (Liu et al 2018a). The DAG’s edges can be pruned via a sparsity regularization term. The optimization objective of DSO-NAS is thus:

Accuracy + L2-regularization(W) + L1-sparsity(\lambda),

where W is the shared weights and \lambda specifies which edges in the DAG are used.

There are 3 phases of optimization:
1. All edges are activated and the shared weights W are trained using normal SGD. Note that this step does not involve \lambda.
2. \lambda is trained using Accelerated Proximal Gradient (APG, Huang and Wang 2018).
3. The best architecture is selected and retrained from scratch.

This procedure works for all architectures and objectives. However, DSO-NAS further proposes to incorporate the computation expense of architectures into step (2) above, leading to their found architectures having fewer parameters and a smaller FLOP counts.

Their experiments confirm all the hypotheses (DSO-NAS can find architectures, having small FLOP counts, having good performances on CIFAR-10 and ImageNet).

Strengths:
1. Regularization by sparsity is a neat idea.

2. The authors claim to be the first NAS algorithm to perform direct search on ImageNet. Honestly, I cannot confirm this claim (not sure if I have seen all NAS papers out there), but if it is the case, then it is impressive.

3. Incorporating architecture costs into the search objective is nice. However, this contribution seems to be orthogonal to the sparsity regularization, which, I suppose, is the main point of the paper.

Weaknesses:
1. Some experimental details are missing. I’m going to list them here:
- Was the auxiliary tower used during the training of the shared weights W?

- Figure 4 does not illustrate M=4 and N=4, e.g. which operation belongs to which layer?

- Did the experiments on CIFAR-10 and ImageNet use the cosine learning rate schedule [1]? If or if not, either way, you should specify it in a revised version of this paper, e.g. did you use the cosine schedule in the first 120 steps to train the shared parameters W, did you use it in the retraining from scratch?

- In Section 3.3, it is written that “The sparse regularization of \lambda induces great difficulties in optimization”. This triggers my curiosity of which difficulty is it? It would be nice to see this point more elaborated, and to see ablation study experiments.

2. Missed citation: MnasNet [2] also incorporates the cost of architectures in their search process. On ImageNet, your performance is similar to theirs. I think this will be a good comparison.

3. The paper has some grammatical errors. I obviously missed many, but here are the one I found:

- Section 3.3: “Different from pruning, which the search space is usually quite limited”. “which” should be “whose”?

- Section 4.4.1: “DSO-NAS can also search architecture [...]”  -> “DSO-NAS can also search for architectures [...]”

References.
[1] SGDR: Stochastic Gradient Descent with Warm Restarts. https://arxiv.org/pdf/1608.03983.pdf

[2] MnasNet: Platform-Aware Neural Architecture Search for Mobile. https://arxiv.org/pdf/1807.11626.pdf

---

> ### Author Response · Authors · 2018-11-26
> **Response**
>
> Thanks for your valuable comments. It helps us to prepare the revision. We address all your concerns in the revision as below.
>
> Q1: Was the auxiliary tower used during the training of the shared weights W?
> A1: Auxiliary tower is used only in the retraining stage.
>
> Q2: “Did the experiments on CIFAR-10 and ImageNet use the cosine learning rate schedule?”
> A2:
> CIFAR: In the pretrain stage and search stage, the learning rate is fixed to 0.1 with batch size 128; In the retraining stage, we use cosine learning rate schedule.
> ImageNet: In the pretrain stage and search stage, the learning rate is fixed to 0.1 with batch 224; In the retraining stage, we use linear decay learning rate schedule.
>
> Q3: “Figure 4 does not illustrate M=4 and N=4, e.g. which operation belongs to which layer?”
> A3: In the revision, we replace the Figure 4 with a new version which has more details. As show in Figure 4, all the operators in level 4 are pruned.
>
> Q4: “The sparse regularization of \lambda induces great difficulties in optimization”
> A4: The non-smooth regularization introduced by l1 regularization makes traditional stochastic SGD failed to yield sparse results. If we need exact zero, we have to use heuristic thresholding on the \lambda learned, which has already been demonstrated in SSS [1] that is inferior. Besides, traditional APG method is not friendly for deep learning as extra forward-backward computation is required, also as shown by SSS.
>
> Q5: “Missed citation: MnasNet also incorporates the cost of architectures in their search process. On ImageNet, your performance is similar to theirs. I think this will be a good comparison.”
> A5: We have added the result of MnasNet [2] in Table 2. Indeed, MnasNet achieves similar results with us with less FLOPs. However, it is also need to note that MnasNet evaluates more than 8K models, which introduces much higher search cost than our method. Moreover, the design space of MnasNet is significant different from other existing NAS methods including ours. It is interesting to explore the combination of MnasNet with ours in the future work.
>
> Q6: “The paper has some grammatical errors.”
> A6: We have fixed the typos and grammatical errors in the revision.
>
> Q7: About “first NAS algorithm to perform direct search on ImageNet”
> A7: We check this claim again and find methods like MnasNet [2] and one-shot architecture search [3] also have the ability to perform direct search on ImageNet, we have delete this claim in the paper. However, to the best of our knowledge, our method is the first method to perform directly search without block structure sharing. We also report preliminary results that directly search on task beyond classification (semantic segmentation). Please refer to Q1 of Reviewer3 for details.
>
> [1] Data-Driven Sparse Structure Selection for Deep Neural Networks. ECCV 2018.
> [2] MnasNet: Platform-Aware Neural Architecture Search for Mobile. https://arxiv.org/pdf/1807.11626.pdf
> [3] Understanding and simplifying one-shot architecture search. ICML 2018.

---

### Public Comment · ~Ludovic_Denoyer1 · 2018-11-06
**Relevant Reference**

Hi,

Here is a relevant reference published at CVPR 2018, with a close idea:  Learning Time/Memory-Efficient Deep Architectures with Budgeted Super Networks -- Tom Veniat, Ludovic Denoyer.

 In that work, edges are pruned by using a budgeted cost directly integrated in the objective function, and optimized through stochastic gradient descent. Could also be used as a comparison.

---

> ### Author Response · Authors · 2018-11-10
> **Reply to "Relevant Reference"**
>
> Thanks for your comment! This paper is indeed very related to our discussion about network structure learning. We will add reference and discussion to it in the revised version.

---

### Public Comment · (anonymous) · 2018-11-06
**Network Pruning is Neural Architecture Search.**

Hi all,

We can understand this paper from another perspective.
As we all know, network pruning (i.e., filter pruning, channel pruning and so on) can be treated as neural architecture search.
Given a pre-trained model which can be treated as a fully-connected DAG, network pruning aims to remove redundant filters and its connections to make DAG sparse.
This paper belongs to fine-grained pruning and the used approach is nearly the same as "Data-Driven Sparse Structure Selection for Deep Neural Networks" which is published at ECCV2018.
That means the same approach, solving the same problem, is submitted to two different communities.

---

> ### Author Response · Authors · 2018-11-07
> **It is just the main contribution of the paper.**
>
> In the front
> "As we all know, network pruning (i.e., filter pruning, channel pruning and so on) can be treated as neural architecture search." If you said "as we all know", it is better to have a reference here. As far as I know, this claim is just one of the ICLR submissions this year: https://openreview.net/forum?id=rJlnB3C5Ym&noteId=SyxGOzHu2Q I think you should comment on their paper with this claim.
>
> First, what you said is just one main contribution of our paper clearly listed in introduction section. The existing NAS methods all start from empty block and then add operators. Our method conveys another totally novel view of NAS that is you can start from full view then prune the useless ones! And more importantly, we prove it works at least comparable or even better than previous approaches! Nobody before us ever think about NAS in this way.
>
> Second, the design space is fundamentally different from model pruning. Pruning from such dense connected block requires deliberate design of the optimization and training scheme. That is why we design three stages training method for DSO-NAS. Moreover, while SSS focuses on pruning the neurons, groups or blocks, our method focuses on pruning the connections between different layers, namely structural connections. The optimization method is indeed from the ECCV paper, but we treat it as an existing, well-developed component to use, and does not declare it as our main contribution.
>
> Above all, we think the justification of "the same approach, solving the same problem, is submitted to two different communities" are arbitrary and unsound.

---

> > ### Public Comment · (anonymous) · 2018-11-07
> > **Response to authors**
> >
> > Thx for your detailed answer. But I still strongly recommend to add some references that use AutoML approaches for model compression.  And in these papers, they indeed explicitly discuss the relationships between network pruning and Neural Architecture Search. I think this is a useful straightforward extension from network pruning by modifying the search space.
> >
> > 1: "AMC: Automl for model compression and acceleration on mobile devices", ECCV2018.
> > 2: "N2N LEARNING: NETWORK TO NETWORK COMPRESSION VIA POLICY GRADIENT REINFORCEMENT LEARNING", ICLR2018.

---

> > > ### Author Response · Authors · 2018-11-08
> > > **Thanks for the constructive suggestions**
> > >
> > > Sure, we will add these references, and discuss the relationships with them in the revision of rebuttal.

---

### Comment · Area_Chair1 · 2018-11-20
**latency**

Latency is a practical measurement of performance. FLOPS is not.

The complicated learned architecture and many branches in Figure 4 make me concerned about the actual latency of the model and the practicality of deploying it, despite the low reported FLOPs.

The authors are encouraged to report the latency in Table 2, and compare it with MnasNet/MobileNet.

---

> ### Author Response · Authors · 2018-11-26
> **Response**
>
> Thanks for your valuable comment. You have indeed raised a very good point to NAS community.
>
> First, it should be noted that the latency of a model is highly dependent on the hardware platform and its corresponding implementation. For example, we test the latency of MobileNet and several state-of-the-art NAS methods, including MnasNet, DARTS, NasNet and our DSO-Nas in different hardware architectures and platforms. During testing, the batch-size is 1 and the input image size is 224 × 224. For GPU testing, a single NVIDIA GeForce GTX 1080Ti is used. The convolution library is CUDNN 7.0. For CPU testing, the test device is Intel i5-6600K CPU. Each network is randomly initialized and evaluated for 500 times. The average runtime is reported.
>
> ---------------------------------------------------------------------------------------------------------------------------
> model                          mxnet(GPU)                       mxnet(CPU)                      TensorRT(GPU)
> ---------------------------------------------------------------------------------------------------------------------------
> MobileNet                         1.94                                   194.18                                 1.01
> ---------------------------------------------------------------------------------------------------------------------------
> MnasNet                          2.85                                     62.32                                 1.71
> ---------------------------------------------------------------------------------------------------------------------------
> DARTS                            6.91                                     64.86                                     -
> ---------------------------------------------------------------------------------------------------------------------------
> NasNet                            9.32                                     92.12                                     -
> ---------------------------------------------------------------------------------------------------------------------------
> DSO-Nas                         7.00                                    149.53                                 4.25
> ---------------------------------------------------------------------------------------------------------------------------
>
> The results test on GPU with MXNet shows that DARTS, NasNet and DSO-Nas have higher latency than MobileNet and MnasNet. This is because the network structures of DARTS, NasNet and DSO-Nas have more fragments than MobileNet and MnasNet due to the unlimited search space. The searched structure of block in DARTS, NasNet and DSO-Nas has a lot of small operators which will reduce degree of parallelism on GPU as shown in ShuffleNetV2 [1]. As for the CPU test results, we found that the latency of MobileNet is much higher, since the memory access is no longer the bottleneck. Compared within NAS method, our method is similar to DARTS, while better than NASNet in terms of accuracy.
>
> When using TensorRT, all the methods benefit from the deliberated implementation in GPU.
>
> Thus, several important factors have considerable affection on latency, including network architectures, hardware architectures and platforms. For our DSO-NAS, we don’t assume any target hardware platform, thus it is hard to directly optimize running latency. In one hand, we could optimize surrogate metric to latency such as MAC as illustrated in [1]; on the other hand, directly optimizing latency is on our schedule for future works as in the conclusion part. We may combine the spirit of MnasNet and our DSO-NAS in one unified framework, however it is out the scope of this single paper.
>
> [1] Ma, N., Zhang, X., Zheng, H.T. and Sun, J., 2018. ShuffleNet v2: Practical guidelines for efficient cnn architecture design. ECCV 2018.

---

> > ### Comment · Area_Chair1 · 2018-11-27
> > **Only reporting FLOPs is misleading**
> >
> > Thanks the authors for the updated results.
> >
> > DSO-NAS is slower than both DARTS and MnasNet, upto 3x slower, although they have similar FLOPs. Only reporting FLOPs is misleading. This validated my concern.
> >
> > The CPU latency for MobileNet looks very slow. I'm concerned if you have turned on MKL-DNN (https://github.com/intel/mkl-dnn) when measuring the CPU speed? MobileNet can easily run below 100ms on a 3 year old Android phone.
> >
> > MobileNet+TF-Lite+Android is a well-established measurement setup for fair comparisons. MobileNet is also designed for mobile platform. The authors are encouraged to perform apple to apple comparison.

---

> > > ### Comment · AnonReviewer1 · 2018-11-27
> > > **Is this the right focus?**
> > >
> > > I'm curious, why is it important to compare the resultant network to MobileNet, when it performs significantly better?
> > >
> > > The additional branches that indeed cause some increase in latency will contribute towards these performance gains, so it's effectively a trade-off.
> > >
> > > I don't want to speak on behalf of the authors, but surely the focus of this paper is their optimization method rather than the latency of the end product.

---

> > > > ### Author Response · Authors · 2018-11-27
> > > > **Response**
> > > >
> > > > Thanks for your clarification. We have updated the results on CPU on the table above. We accidentally report swap our result and NASNet. The results are more reasonable on CPU now.

---

> > > > ### Comment · Area_Chair1 · 2018-12-09
> > > > **compare with MnasNet**
> > > >
> > > > Should compare with MnasNet.
> > > > This work is both slower and less accurate than the existing mnasNet. From a practical deployment perspective, reporting a low FLOP number is not the correct evaluation metric.

---

> > > > > ### Author Response · Authors · 2018-12-10
> > > > > **Response**
> > > > >
> > > > > We have added the MNasNet -92 (without SE) results to Table2 for comparison. Please note that, the difference on Imagenet is only 0.2% in terms of accuracy, which is quite minor compared to the error rate 25.2%. We indeed don't optimize latency intentionally in this work, however this should not be hard if we could directly test the latency on target hardware.
> > > > >
> > > > > Another noteworthy point is that MNasNet does not report their search cost in their paper. As noted in a recent work (https://arxiv.org/abs/1812.00332), MNasNet needs about 10^4 GPU hours for search! In contrast, we only need 6 GPU hours. The difference here is more than 1000 times! If you concern more about practical use, we think the cost of searching the model is also an important factor for practical use in NAS. We would like to ask the area chair to consider this point in the decision.

---

> > > ### Author Response · Authors · 2018-11-27
> > > **Updated results on CPU**
> > >
> > > Thanks for your response. We double check the previous results, finding that we accidently swap the results of NasNet and DSO-Nas on mxnet cpu test in the Table we reported before. Thus, our method is only 25% slower than DARTS. To make a more complete comparison, we turn on MKL-DNN and update our cpu results. The test device is Intel Xeon CPU E5-2670 v3. The platform is MXNet.
> > >  ---------------------------------------------------------------------------------------------------------------------------
> > > model           mxnet(GPU)         mxnet(CPU)         mxnet(CPU+mkldnn)       TensorRT(GPU)
> > > ---------------------------------------------------------------------------------------------------------------------------
> > > MobileNet           1.94                    280.66                       26.51                              1.01
> > > ---------------------------------------------------------------------------------------------------------------------------
> > > MnasNet             2.85                      73.75                        7.14                               1.71
> > > ---------------------------------------------------------------------------------------------------------------------------
> > > DARTS               6.91                       83.74                      17.00                                 -
> > > ---------------------------------------------------------------------------------------------------------------------------
> > > NasNet               9.32                     198.47                      30.58                                 -
> > > ---------------------------------------------------------------------------------------------------------------------------
> > > DSO-Nas           7.00                      111.38                      22.34                              4.25
> > > ---------------------------------------------------------------------------------------------------------------------------
> > >
> > > The results here are more reasonable in the updated version. The trends are similar with and without MKLDNN in CPU. In particular, our method is slightly faster than MobileNet, 25% slower than DARTS, and about 30% faster than NasNet. Note that we haven’t specifically optimize latency as indicated in paper. This is definitely a promising direction to pursue in future work.

---

### Meta-Review · Area_Chair1 · 2018-12-17
**borderline**

**Confidence:** 3
**Recommendation:** Reject

**Metareview:**

This paper proposes Direct Sparse Optimization (DSO)-NAS to obtain neural architectures on specific problems at a reasonable computational cost. Regularization by sparsity is a neat idea, but similar idea has been discussed by many pruning papers. "model pruning formulation for neural architecture search based on sparse optimization" is claimed to be the main contribution, but it's debatable if such contribution is strong: worse accuracy, more computation, more #parameters than Mnas (less search time, but also worse search quality). The effect of each proposed technique is appropriately evaluated. However, the reviewers are concerned that the proposed method does not outperform the existing state-of-the-art methods in terms of classification accuracy. There's also some concerns about the search space of the proposed method. It is debatable about claim that "the first NAS algorithm to perform direct search on ImageNet" and "the first method to perform direct search without block structure sharing". Given the acceptance rate of ICLR should be <30%, I would say this paper is good but not outstanding.

---

> ### Author Response · Authors · 2018-12-21
> **...**
>
> We regret the decision is so arbitrary and unconvincing, even AC cannot foresee the scientific value of the work, only concern the comparison of the results.